# Do mobile device apps designed to support medication adherence demonstrate efficacy? A systematic review of randomised controlled trials, with meta-analysis

Laura Catherine Armitage [1,2] Aikaterini Kassavou [2] Stephen Sutton[2]

¹Nuffield Department of Primary Care Health Sciences, University of Oxford, Oxford, UK
²Department of Public Health and Primary Care, University of Cambridge, Cambridge, UK

**Correspondence to**
Dr Laura Catherine Armitage;
laura.armitage@phc.ox.ac.uk

## ABSTRACT

**Objectives** To estimate the efficacy of app-based interventions designed to support medication adherence and investigate which behaviour change techniques (BCTs) used by the apps are associated with efficacy.

**Design** Systematic review of randomised controlled trials (RCTs), with meta-analysis.

**Setting** Medline/PubMed, PsycINFO, Cumulative Index to Nursing and Allied Health Literature, Embase and Web of Science were searched from 1990 to November 2018 for RCTs conducted in any healthcare setting.

**Participants** Studies of participants of any age taking prescribed medication for any health condition and for any duration.

**Intervention** An app-based intervention delivered through a smartphone, tablet computer or personal digital assistant to help, support or advise about medication adherence.

**Comparator** One of (1) usual care, (2) a control app which did not use any BCTs to improve medication adherence or (3) a non-app-based comparator.

**Primary and secondary outcome measures** The primary outcome was the pooled effect size of changes in medication adherence. The secondary outcome was the association between BCTs used by the apps and the effect size.

**Results** The initial search identified 13 259 citations. After title and abstract screening, full-text articles of 83 studies were screened for eligibility. Nine RCTs with 1159 recruited participants were included. The mean age of participants was >50 years in all but one study. Health conditions of target populations included cardiovascular disease, depression, Parkinson's disease, psoriasis and multimorbidity. The meta-analysis indicated that patients who use mobile apps to support them in taking medications are more likely to self-report adherence to medications (OR 2.120, 95% CI 1.635 to 2.747, n=988) than those in the comparator groups. Meta-regression of the BCTs did not reveal any significant associations with effect size.

**Conclusions** App-based medication adherence interventions may have a positive effect on patient adherence. Larger scale studies are required to further evaluate this effect, including long-term sustainability, and intervention and participant characteristics that are associated with efficacy and app usage.

## Strengths and limitations of this study

► This systematic review and meta-analysis of randomised controlled trials offers new information on the efficacy of mobile device apps in supporting medication adherence across a range of medical conditions.

► This review is reported according to the Preferred Reporting Items for Systematic Reviews and Meta-Analyses statement.

► A sensitive search strategy was designed, and screening of articles and data extraction were performed independently by one clinical researcher and one non-clinical researcher.

► The review protocol was prospectively registered, and the scope of the review, inclusion and exclusion criteria remained unchanged throughout.

► Two reviewers independently considered the abstracts and full-text articles for inclusion, extracted the data, and assessed the included studies for risk of bias in the same way.

**Prospero registration number** PROSPERO Protocol Registration Number: CRD42017080150.

## INTRODUCTION

Adherence to medication is defined as the extent to which patients take medications as prescribed and agreed with their healthcare providers.[1] The therapeutic benefit of prescribed medications is limited owing to an estimated 50% of patients not adhering to medications.[2] The annual cost of non-adherence is estimated to exceed £930 million in England[3] and between $100 and $300 billion in the USA.[4] Furthermore, in 2003 the Adherence to Long-term Therapies report published by the WHO stated that unless medication adherence is addressed, advances in biomedical technology will not achieve their potential.[5]

Interventions for improving adherence should be reliable, contemporary, acceptable and readily available to the person. Mobile devices can meet these needs, and in 2018 Ofcom reported that 78% of adults and 95% of individuals aged 16–24 years old in the UK own a smartphone, demonstrating that smartphones are a growing part of modern life.[6] Patients who are prescribed multiple medications are commonly in the older age group, where the percentage of smartphone users may be lower. However, smartphone ownership in those aged over 65 more than trebled between the years 2012 and 2015 from 5% to 18%, and is set to continue to increase.[7]

Hundreds of apps are now available to patients to support them in taking regular medication.[8] Such apps frequently use behaviour change techniques (BCTs) to promote improvements in adherence. A BCT is defined as an active component of an intervention that is designed to change behaviour.[9] To date, the efficacy of smartphone apps in supporting patient adherence to regularly prescribed medications has not been evaluated within the scientific literature using rigorous quantitative review. We have performed a novel systematic review with meta-analysis to evaluate the efficacy of smartphone apps in supporting medication adherence.

The objectives of this review are (1) to establish whether apps designed to support medication adherence demonstrate efficacy and (2) to identify the intervention characteristics and BCTs associated with efficacy. The conclusions drawn have potential to inform future large-scale public health interventions for the improvement of medication adherence.

## METHODS
This systematic review is reported according to the Preferred Reporting Items for Systematic Reviews and Meta-Analyses statement.[10] The protocol for this review is registered with and published on the PROSPERO (International Prospective Register of Systematic Reviews) database of systematic reviews.[11] The scope of the review, inclusion and exclusion criteria remained unchanged throughout.

### Patient and public involvement
This research was done without patient involvement. Patients were not invited to comment on the study design and were not consulted to develop patient-relevant outcomes or interpret the results. Patients were not invited to contribute to the writing or editing of this document for readability or accuracy.

### Data sources
Medline/PubMed, PsycINFO, Cumulative Index to Nursing and Allied Health Literature, Embase and Web of Science electronic databases were searched from 1990 to November 2018. A sensitive search strategy was designed based on preliminary searches, relevant papers and keywords. Key search terms included adherence,

non-adherence, smartphone app and randomised controlled trials (RCTs). An example detailed search strategy for the Medline/PubMed database can be found in online supplementary etext 1. The reference lists of the included studies were screened for additional relevant trials.

Two reviewers (LCA and AK) independently screened all citations by title and abstract and excluded those which were irrelevant or clearly did not meet the eligibility criteria. Full-text articles were obtained for all remaining citations and screened against the eligibility criteria by the same reviewers. Any discrepancies at any stage in the screening process were settled through discussion between LCA, AK and a third reviewer (SS).

### Study selection
RCTs relevant to this review were those that investigated the use of an app on a mobile device to support medication adherence. The inclusion criteria were as follows:
► Participants of any age who were taking one or more prescribed medications for any health condition and for any duration.
► An intervention group which received an app-based intervention delivered through a smartphone, tablet computer or personal digital assistant to help, support or advise about medication adherence.
► A comparator group which received usual care, a control condition which did not use an app[12] or a control app which did not include any BCTs (eg, an app that was a non-health-related game[13]).
► Medication adherence data were reported for both the intervention and comparator groups.
  The exclusion criteria were as follows:
► Complex interventions of which an app was just one component.
► Interventions delivered by text messaging or telemedicine.
► Outcome assessed was adherence to treatment components other than prescribed medication. For example, one study reported outcome data as a composite score, combining adherence to other aspects of the medical regimen (such as clinic attendance and laboratory work) with adherence to medication.[14]
► A study population of healthy recruits who were required to adhere to a placebo medication.
► Comparator group received an adherence intervention in another format, such as a different medication adherence app.

Published articles were required to be in the English language.

### Data extraction and synthesis
Data were extracted independently by two reviewers (LCA and AK) using a custom data extraction form. Participants' baseline characteristics were extracted, including age and sex. Raw data on medication adherence were extracted either from the published articles or by contacting the authors of the primary studies by

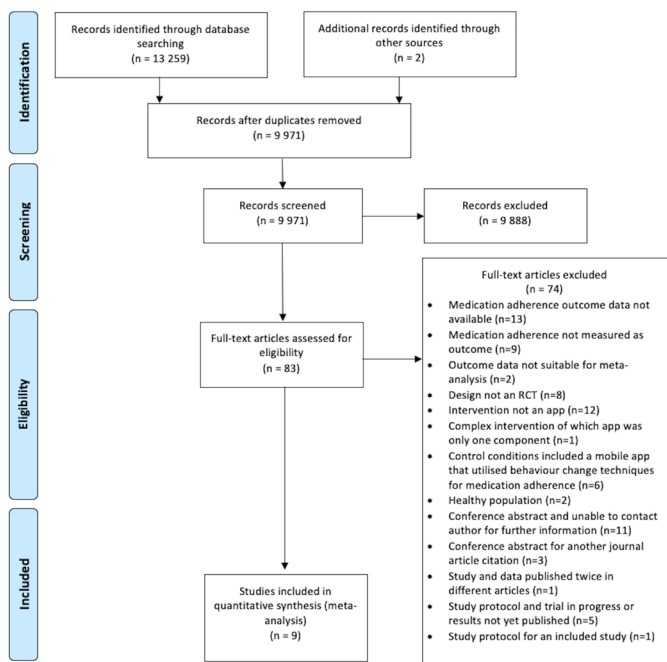

**Figure 1** PRISMA flow chart. PRISMA, Preferred Reporting Items for Systematic Reviews and Meta-Analyses; RCT, randomised controlled trial.

email. When two measures of adherence were reported, that which was considered most comparable with the outcome measures of the other included studies was used for meta-analysis. For example, one study reported adherence according to both the number of times an electronically chipped dispenser was actioned by the patient and the weight of the dispensing canister at study follow-up, and the reviewers agreed that the action of dispensing was more comparable with that of the other included studies.[15] Authors were contacted by email for any other data required which were not available in the published articles. Where more than one app intervention was tested, data for the intervention considered most comparable with the other studies were extracted for meta-analysis. For example, one study investigated a 'basic app' (which provided a unidirectional once-daily reminder) as well as an 'advanced app' (which provided customisable and interactive features) versus usual care.[16] The reviewers agreed that the advanced app was most comparable with the interventions of the other studies and so data pertaining to this intervention were extracted. BCTs that were used in the app-based interventions were coded according to the Behaviour Change Technique Taxonomy.[17] This is a structured taxonomy of techniques used by interventions to change behaviour. The BCT taxonomy does not have a code relating to performance of behaviour, and therefore where participants were required to enter whether or not medication was taken, this was coded as 'Report whether or not the behaviour was performed'. Similarly, the taxonomy does not have a code relating to tailoring of an intervention, and therefore where a study used information gained about a

person to define the content, frequency or format of an intervention, this was coded as a 'Tailored' intervention.[18]

The included trials were assessed for risk of bias using the Cochrane Risk of Bias Tool.[19] Two reviewers (LCA and AK) individually assessed each of the included articles against the criteria outlined in the tool. Any discrepancies at any stage in the coding process were settled through discussion between the three reviewers (LCA, AK, SS). Studies were considered to be at unclear risk of bias for an item when the primary article did not provide sufficient information for the reviewers to decide categorically between high or low risk of bias.

### Statistical analysis
Outcome data on medication adherence were extracted from each of the included studies at the final observation point. Authors were contacted for raw final observation point adherence data if these were not reported in published manuscripts. Trials were considered homogeneous based on the content of the intervention and comparator groups and the outcome measurement of medication adherence. The Comprehensive Meta-Analysis Software[20] was used to estimate the overall effect of app-delivered interventions on medication adherence using a random-effects model and to perform meta-regression using the BCTs used by the apps. The Comprehensive Meta-Analysis Software conversion function was used where required to convert extracted raw data into OR and standardised mean difference between the intervention and control groups.[21] The $I^2$ statistics were used to estimate heterogeneity between the included studies.

### RESULTS
The initial electronic database search identified 13 259 potentially eligible studies, and 2 were identified from the reference lists of identified articles. After removal of duplicates, 9971 citations were screened by title and abstract. We considered 83 of these to be eligible and retrieved the full-text articles. Of these, nine studies reported outcome data eligible for meta-analysis.[12 13 15 16 22–26] The screening process and reasons for exclusion are reported in figure 1.

The characteristics of the nine studies are shown in table 1. Publication year of the included studies ranged from 2014 to 2018. The median sample size was 102 participants (range 24–412), and the median follow-up period was 12 weeks (range 28 days–16 weeks).

The mean age of participants in the intervention group was reported by eight studies and ranged from 20.3 (SD 4.0)[22] to 73.8 (SD 8)[12]; the mean of the mean ages across these eight studies was 56.7 years. One study did not report the mean age but presented the percentage of participants whose age fell within a series of bandwidths.[15] The percentage of participants in the intervention groups who were female ranged from 11%[16] to 89%,[22] and the mean percentage of female participants among the included studies was 45.2%.

**Table 1** Study characteristics

| Source (sample size) | Country of study | Study design | Health problem of target population | Recruitment setting | Participant characteristics at recruitment | | | | App intervention | Outcome measure | Follow-up interval |
|---|---|---|---|---|---|---|---|---|---|---|---|
| | | | | | Intervention group | | Comparator group | | | | |
| | | | | | Mean age (SD) | Female, n (%) | Mean age (SD) | Female, n (%) | | | |
| Hammonds et al[22] (57) | USA | RCT | Depression | College students were recruited via campus-based advertisements and the university research recruitment system. | 20.3 (4.0) | 25 (83) | 20.9 (4.7) | 24 (89) | Medication reminder app. | Number of participants who have taken ≥80% prescribed tablets per day. | 25–35 days |
| Labovitz et al[23] (28) | USA | RCT | Ischaemic stroke requiring secondary oral anticoagulation | Outpatient clinic. | 58.3 (9.79) | 6 (40) | 55.5 (16.55) | 9 (69) | Artificial intelligence app. | Pill count. | 12 weeks |
| Lakshminarayana et al[24] (215) | UK | RCT | Parkinson's disease | Outpatient clinic. | 59.86 (9.13) | 34 (36.2) | 60.71 (10.26) | 45 (42.1) | Parkinson's tracker app. | Self-report: 8-item MMAS. | 16 weeks |
| Mertens et al[12] (24) | Germany | RCT crossover | Cardiovascular disease | Cardiac rehabilitation groups. | 73.8* | 12 (50)* | 73.8* | 12 (50)* | Medication plan app (V.1.3). | Self-report: A14 scale. | 28 days |
| Mira et al[25] (102) | Spain | RCT | Multicomorbidity | Community health centres. | 70.9 (8) | 21 (41) | 72.9 (6) | 23 (48) | ALICE medication self-management app. | Self-report: 4-item MMAS. | 3 months |
| Morawski et al[26] (412) | USA | RCT | Hypertension | Social media, mobile apps, targeted advertisements and online communications. | 51.7 (10.5) | 120 (57.4) | 52.4 (10.1) | 127 (62.9) | MediSAFE-BP app. | Self-report: 8-item MMAS. | 12 weeks |
| Santo et al[16] (163) | Australia | RCT | Coronary heart disease | Inpatient, outpatient and rehabilitative care settings. | 58.4 (9.04) | 14 (13) | 56.8 (8.64) | 6 (11) | MedApp-CHD. | Self-report: 8-item MMAS. | 3 months |
| Shah et al[13] (24) | USA | RCT | Cardiovascular stent | Patients in the immediate postoperative phase of insertion of a drug-eluting cardiovascular stent. | 60.5 (8.9) | 6 (46) | 60.6 (4.1) | 4 (36) | MyIDEA education app. | Medication possession ratio and self-report: 8-item MMAS. | 90 days |

Continued

**Table 1** Continued

| Source (sample size) | Country of study | Study design | Health problem of target population | Recruitment setting | Participant characteristics at recruitment | | | | App intervention | Outcome measure | Follow-up interval |
|---|---|---|---|---|---|---|---|---|---|---|---|
| | | | | | Intervention group | | Comparator group | | | | |
| | | | | | Mean age (SD) | Female, n (%) | Mean age (SD) | Female, n (%) | | | |
| Svendsen et al[15] (134) | Denmark | RCT | Psoriasis | Hospital posters and the dermatology outpatient clinic. | 18–40 29†<br>41–50 21<br>51–60 26<br>61–75 24 | 27 (40) | 18–40 32†<br>41–50 20<br>51–60 24<br>61–75 24 | 25 (38) | MyPso SmarTop V.1.0 or SmarTop number 0536776. | Measured via a digital chip record in the medication dispenser. | 28 days |

*Baseline characteristics presented for the whole cohort as study design was a randomised cross-over trial.
†Number of participants in each age group (years) presented as mean age not available.
MMAS, Morisky Medication Adherence Scale; RCT, randomised controlled trial.

Health problems of the target population varied between studies. Five studies sampled a patient population with cardiovascular disease,[12 13 16 23 26] and each of the remaining four studies enrolled patient populations with different health problems: depression,[22] Parkinson's disease,[24] psoriasis[15] and multimorbidity.[25]

### Outcome measurement methods

A number of measures of medication adherence were used in the included studies. Of the nine studies, four measured adherence using the eight-item Morisky Medication Adherence Scale,[24 26 27] with one of these also measuring adherence through a medication possession ratio (see online supplementary material).[13] For this study, the Morisky Medication Adherence Scale data were extracted for the meta-analysis as these were most comparable with the data available for other studies. A further study measured adherence using the four-item Morisky Medication Adherence Scale.[25] Of the remaining studies, two reported adherence according to pill count,[22 23] one the A14 self-report scale of medication adherence[12] and one through a digital chip record fitted within the medication dispenser (table 1).[15]

### Meta-analysis

Meta-analysis was performed, pooling results from the nine eligible studies which reported data on 988 participants, to estimate the effect of mobile apps in improving medication adherence.[12 13 15 16 22–26] The results of this meta-analysis indicated that patients who participated in medication adherence interventions delivered by mobile apps were more likely to adhere to prescribed medications (OR 2.120, 95% CI 1.635 to 2.747) than those who did not use such interventions. The $I^2$ value was 9.8, indicating low statistical heterogeneity between the studies included in the meta-analysis. The forest plot of the meta-analysis is shown in figure 2. The weighting of each individual study in the meta-analysis can be seen in online supplementary etable 1, and raw data extracted for the meta-analysis can be found in online supplementary etable 2.

A focused meta-analysis was performed, pooling results from the five studies which reported medication adherence according to the Morisky Medication Adherence Scale.[13 16 24–26] The results of this also indicated that patients using medication adherence interventions delivered by mobile apps reported higher adherence to prescribed medications (OR 1.83, 95% CI 1.42 to 2.36).

### Behaviour change techniques

The most common BCT used by apps was 'Tailored', which was used by all studies. The next most common was 'Prompts and Cues', used by seven studies,[12 15 16 22–25] and 'Report whether or not the behaviour was performed', which was used by six studies.[12 16 22–25] Information on the BCTs coded for each of the interventions and the definitions of each of the BCTs are available in online supplementary etable 3 and 4. Agreement between reviewers for classification of BCTs used by the interventions was

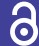

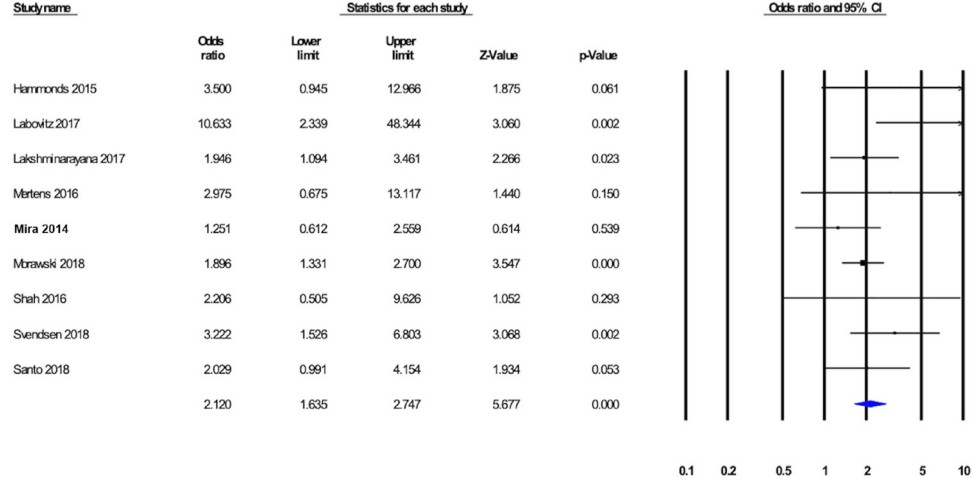

**Figure 2** Forest plot of randomised trials evaluating a mobile device app to support medication adherence against comparators.

assessed using the kappa statistics. This revealed a 'very good' level of agreement (kappa=0.902, 95% CI 0.835 to 0.969). All BCTs coded for each of the study interventions are provided in online supplementary etable 3. Those coded in more than three but less than six of the nine studies were included in the meta-regression; this did not reveal any significant associations between the BCTs used and the effect size.

### Risk of bias at primary study level

A summary of the risk of bias analysis is shown in table 2. Overall, all studies were deemed as unclear in their risk of bias.

The most common source of risk of bias among the studies was the absence of blinding of participants and personnel to the outcome measure. The second most common risk of bias arose from a lack of objective measurements of medication adherence. This was due to the use of self-report measures of adherence: six of the studies included in the meta-analysis reported adherence based on a self-assessment scale,[12 13 16 24–26] and one study each used patient-entered medication logs in an app,[22] an electronically chipped medicated foam dispenser[15] and pill counts at clinic visits.[23]

Self-assessment scales are a form of self-report and therefore a source of potential bias, including those of social desirability, and those arising from selective recall and the duration of the recall period.[28] Two studies included in this review which measured adherence through participant self-report, sought to compare their subjective measurements of adherence with serum measurements of either drug-reactivity units or a medication's anticoagulant effect.[13 23] One study showed that serum measurement of the medication's anticoagulant effect was similar between both the intervention and comparator groups, despite adherence measures being better in the intervention group.[23] The other study revealed that drug-reactivity

units were worse in the intervention group than in the control group, which conflicted with measured adherence outcome data.[13] However, the presence of a drug biomarker does not equate to compliance and the absence does not equate to non-compliance.[29] Indeed, drug metabolism should be taken into account for such measures of medication adherence, as physiological state and metabolic rate vary among individuals.[30]

### Publication bias

A funnel plot of effect size estimates of each of the studies included in the meta-analysis is provided in online supplementary efigure 1. Formal analysis of plot symmetry was not performed owing to the total number of studies included being less than 10.[31] Visual inspection reveals moderate symmetry of the plot.

### DISCUSSION
### Principal findings

We conducted a rigorous systematic review of the literature to investigate whether mobile device apps demonstrate efficacy in supporting medication adherence. A random-effects meta-analysis of the pooled results from all nine studies indicated that people who use medication reminder apps are significantly more likely to adhere to their medications than those who do not, but the results should be interpreted with caution owing to the data for six of the nine studies included in the meta-analysis being based on self-reported measures of adherence.

Intervention effects reported between the studies varied, as shown in the forest plot (figure 2). All nine studies reported that their mobile app-delivered interventions demonstrated efficacy in supporting medication adherence, although in five studies the interventions did not have a statistically significant effect.[12 13 16 22 25] Although the $I^2$ measure of statistical heterogeneity was low, and

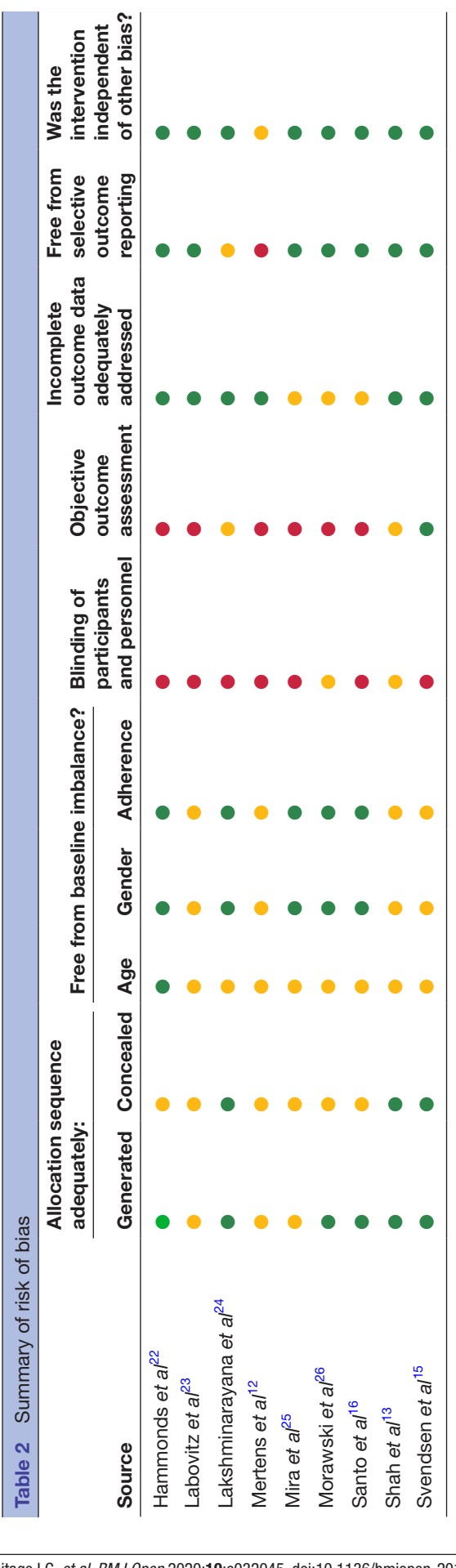

**Table 2** Summary of risk of bias

●: low risk of bias; ●: unclear risk of bias; ●: high risk of bias.

meta-regression of the BCTs used by the studies was not able to account for variance in effect observed between studies, there are other potential sources of methodological and clinical heterogeneity which may account for this. One potential explanation might be the variation in adherence measurement. This may be illustrated by three studies which incorporated five of the same BCTs,[23–25] but where one[23] reported notably better medication adherence (OR 10.633, 95% CI 2.339 to 48.344) than the other two (OR 1.946, 95% CI 1.094 to 3.461; OR 1.251, 95% CI 0.612 to 2.559).[24 25] In Labovitz et al's[23] study, medication adherence was measured by pill counts at clinic follow-up visits, whereas in the other two studies adherence was measured by retrospective self-scoring on a medication adherence scale.[24 25] It could be argued that both of these different measures of adherence are vulnerable to bias, with pill counts potentially being vulnerable to expectation bias and retrospective self-scores being vulnerable to recall bias. We subsequently performed a focused meta-analysis of the five studies which measured adherence using the Morisky Medication Adherence Scale to investigate whether this influenced the effect size observed. While the result of the meta-analysis remained significant, the effect size was lower. This may support the use of real-time or prospective adherence monitoring over retrospective self-report for future research in the field of medication adherence.

A further possible explanation for variance in effect size might be the techniques, beyond those reported in the BCT taxonomy, used by the apps. For example, in the study by Labovitz et al,[23] the app intervention employed artificial intelligence to identify the participant, their medication and ingestion of the medication, using the camera of the mobile device and software algorithms. This could be regarded as a highly tailored intervention, and one might hypothesise that the degree of tailoring could influence the effect size observed. A further example of a highly tailored intervention was seen in the study by Lakshminarayana et al.[24] All of the app-based interventions in the included studies were tailored to the prescribed medication regimen, but in this instance the tailoring went beyond the prescribed medication regimen, to include tailoring to the patients' intentional and non-intentional non-adherence, beliefs about taking medications, mood, cognitive impairment, symptom control, quality of life, age and disease duration. A significant effect size was observed in this study, and this may support a hypothesis that highly tailored apps are more effective at supporting medication adherence than minimally tailored apps.

Further participant-related factors which may have contributed to variance in the effect size seen between studies, include disease burden, non-adherence behaviours, perceived benefit from the medication and any side effects. Interpretation of the overall result of the meta-analysis and generalisability to the wider population should therefore be cautious.



## Strengths and weaknesses of the primary studies

Based on the assessment of the risk of bias undertaken as part of this review, the evidence included in this meta-analysis is of moderate quality. The most common weaknesses in this evidence are small sample size, lack of objectivity of the outcome measurement, lack of blinding of participants and personnel, and limited follow-up time.

## Strengths and weaknesses at review level

Our comprehensive literature search and search of the PROSPERO database indicate this is the first systematic review and meta-analysis to investigate the efficacy of mobile apps in supporting medication adherence. This offers new knowledge to the field; however, as such there are no comparable studies with which the results of this review can be compared.

The pooling of data from a group of studies which used a variety of methods for measurement of medication adherence and targeted at patients with different healthcare problems is a limitation of this review. As further studies in this field are published, it will become more feasible to conduct subgroup analysis or meta-regression analyses against variables such as age, health condition, targeted medication, intervention content and app characteristics. A standardised protocol for measuring adherence and reporting the content and characteristics of app interventions may facilitate further analysis of the latter two variables.

This review cannot report on the likely sustainability of the effect on medication adherence given the short study duration of the studies included in this review; across the nine studies, the maximum follow-up period was 16 weeks. Furthermore, the optimal frequency and duration of app use that are required to achieve improvements in medication adherence are not known. Long-term and large-scale research is required in this field to address the question regarding the sustainability of the observed effect found in this meta-analysis.

This systematic review and meta-analysis included studies with a range of ages of participants. The lowest mean age of participants in the included studies was 20.9 years and the highest was 70.9 years. The ubiquity and use of mobile devices vary across age groups,[7] and while the majority of participants in these studies were aged 50 and over further research to establish the acceptability and usability of adherence apps among different age groups of adults would be valuable.

Finally, the reviewers set out to perform a meta-regression of the BCTs used by app interventions in the included studies.[11] However, this did not reveal any significant association between the BCTs used and the observed effect size. This may be owing to the low estimate of heterogeneity and the small number of eligible included studies.

## CONCLUSIONS

This meta-analysis indicates that medication adherence interventions delivered by smartphone apps are associated with higher levels of self-reported adherence to prescribed medications. However, when considering whether to recommend an app to patients, these results should be interpreted with caution owing to variance in effect size seen between the studies, the small sample sizes taken from different patient populations, the heterogeneity among included studies and the prominence of self-reported scales to measure medication adherence. Further research should aim to first establish a standard protocol for measuring and reporting of adherence in the context of evaluating digital interventions. Subsequently, further research investigating intervention characteristics associated with an app's effect and the sustainability of any effect gained would add to this field.

**Contributors** All authors were involved in the design of the review and undertook methodological planning. LCA performed and refined the searches in consultation with a medical librarian. LCA and AK performed initial screening and data extraction, and SS gave screening advice where any disagreements arose. All authors contributed to performing the meta-analysis and data interpretation. LCA led the writing, and all authors contributed to successive drafts and approved the final manuscript.

**Funding** This paper presents an independent research funded by the National Institute for Health Research (NIHR) under its Programme Grants for Applied Research (grant reference number RP-PG-0615–20013). The views expressed are those of the authors and not necessarily those of the NIHR or the Department of Health and Social Care. NIHR's Collaboration for Leadership in Applied Health Research and Care (CLAHRC) in East of England provided salary support to LCA during the conduct of this review.

**Disclaimer** The funders had no role in review design, data collection, data analysis, data interpretation, writing of the manuscript and decision to submit the manuscript for publication.

**Competing interests** None declared.

**Patient consent for publication** Not required.

**Provenance and peer review** Not commissioned; externally peer reviewed.

**Data availability statement** All data relevant to the study are included in the article or uploaded as supplementary information.

**ORCID iDs**
Laura Catherine Armitage http://orcid.org/0000-0002-5009-4899
Aikaterini Kassavou http://orcid.org/0000-0002-6562-4143

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
