## [Reviewer comments · BMJ Open]

ARTICLE DETAILS

TITLE (PROVISIONAL)	Do mobile device apps designed to support medication adherence demonstrate efficacy? A systematic review of randomised controlled trials, with meta-analysis
AUTHORS	Armitage, Laura; Kassavou, Aikaterini; Sutton, Stephen

VERSION 1 – REVIEW

REVIEWER	Lauri Linder University of Utah United States of America
REVIEW RETURNED	21-Jun-2019

GENERAL COMMENTS	Thank you for the opportunity to review this manuscript. As apps to support health behaviors such as medication adherence become more ubiquitous, the need to evaluate their efficacy is relevant. Given the heterogeneity of the studies in relation to the study sample size and characteristics, approach measuring adherence, and follow up period, I do have some cautions regarding the strength of the claims that can be made regarding the efficacy of apps. In the methods section, please address the inclusion/exclusion criteria earlier within this section and state these criteria more explicitly, including if you sought to include published articles only. The inclusion criteria speak to inclusion of individuals of any age who were taking one or more prescribed medications. Did these prescribed medications need to be taken on a given schedule or were “as needed” prescribed medications regarded as meeting this eligibility criterion? If so, did the studies adequately account for adherence with “as needed” medications in addition to those scheduled, prescribed medications? How, specifically, did you define “app-based intervention” for the sake of your review? Is an app that delivers an alert one that meets criteria? In the discussion section, you speak of patients who “participate in medication adherence interventions delivered by mobile applications.” Please provide a clear conceptual definition of this inclusion criterion and how you operationalized it across studies deemed as meeting criteria. I am also unclear as to how “behavior change technique” was defined for the sake of this review. Again, is a simple reminder alert system sufficient to meet this criterion? Please state clearly. I also do not see sufficient evidence from the manuscript to feel confident that any of these interventions were “tailored.” The NCI defines a tailored intervention as “The use of communication, drugs, or other types of treatments that are specific
--

	for an individual or a group to improve health or change behavior.” If some of the interventions are ones that meet the generally accepted definition of a tailored intervention, please include sufficient supporting evidence to support this claim. Perhaps Table 1 could be expanded to include these descriptions. The first paragraph of the discussion seems better suited to the results section. As I stated earlier, given the heterogeneity of the studies included in this review, I am not fully confident that the authors can claim that those individuals using medication reminder apps are more likely to adhere to their medications. Perhaps the better claim is to indicate that they report higher adherence given the predominance of self-report vs. objective measures to ascertain adherence within this group of studies. General comments:  • Please avoid phrases such as “To our knowledge, this is the first ...,” just state what the project adds to the larger body of knowledge. • In the final paragraph on page 8, lines 52-59, please review how you are using the terms “measured” and “measuring.” The studies are measuring adherence rather than the scale
--	---

REVIEWER	leslie saxon University of Southern California USA
REVIEW RETURNED	16-Jul-2019

GENERAL COMMENTS	In this study, based on a thorough literature review spanning 4 years, nine randomized studies of medication adherence, that utilized various app-based methods for tracking adherence were combined using meta-analysis to determine effectiveness. The study found that Apps were 2 times more likely to drive adherence, compared to control groups. In the introduction, the authors appropriately note the adverse effects on disease outcomes as well as the economic burden of medication non-adherence in patients across the spectrum of chronic disease. They also note that despite the presence of thousands of medication adherence apps designed all over the globe, there has been very little rigorous study of their effectiveness. Also, the authors had to review 9000 studies to find 9 in the published literature that conformed to these criteria. Of the nine studies reported, most assessed adherence using self-reporting, without clear validation of drug ingestion. Nonetheless, the analysis points to the importance and promise of providing comprehensive tools for patient self-management of chronic conditions. Drug adherence is just one aspect of digitally enabled patient self-management. Education and diagnostic tools are another. Medical care communication streams, medication access, nimble virtual care paradigms that don't require a patient and physician to be in the same room are also critical. I think that the strength of this analysis is that it points to the need for more rigorous and comprehensive study medical applications for disease management. Outcome data is critical to identifying key elements of digital management that can scale across disease states and across the globe. Notes for Specific Edits
---

	1. The paper needs to be shortened, both in terms of tables and text. 2. The authors should review behavioral aspects of top-rated drug adherence apps from the app stores, this is important because these are the apps that patients are using 3. The authors should review FDA approved commercial digital pharma products aimed at adherence, like Proteus Digital Health.
--	---

REVIEWER	Karla Santo Hospital Israelita Albert Einstein, São Paulo, Brazil
REVIEW RETURNED	31-Jul-2019

GENERAL COMMENTS	The authors describe a systematic review with meta-analysis that aimed to address an important research question of whether mobile apps are effective in improving medication adherence. In addition, the authors aimed to identify the apps characteristics, its relationship with behavioural change techniques and whether those influenced effectiveness. As mobile apps have been increasingly used in healthcare and, more recently, as a potential tool to improve adherence, this review provides relevant data to support the use of such apps in clinical practice. The paper is well-written, and the review was performed in a proper scientific manner according to the PRISMA guidelines. Of note, the review was registered in the PROSPERO register, had a robust search strategy, and had 2 reviewers independently reviewing the abstracts and full text articles, extracting the data and assessing the included articles for the risk of bias. However, there are concerns about the results presented and a few points in the methods that need clarification, as listed below. 1) Methods - Study selection: In the eligibility criteria, it is stated that the study needed to have an intervention group which received an app-based intervention to help, support or advise about medication adherence. It is unclear whether studies that have a mobile app as one component of a broader intervention addressing medication adherence could be included in the review, for example an intervention that included nurse-led counselling about medication adherence in addition to the mobile app, or whether the mobile app should be the only component of the intervention. Please clarify in the text. 2) Results: a) There are inconsistencies in the information presented in the text and in the flowchart. In the text, it is reported that the database search identified 13,295 articles; however, in the flowchart, it is stated that 13,259 records were identified through database searching. Similarly, in the text, it is reported that 80 articles were eligible for full-text review; however, in the flowchart, it is stated that 83 full-text articles were assessed for eligibility. Please report the correct numbers in the text. b) There also several inconsistencies between the information in the text and in Table 1. For example, in Table 1, the mean age ranged from 20.3 (Hammonds et al, 2015) to 73.8 (Mertens et al, 2016), instead of 20.3 to 70.9 as presented in the text (lines 42-44, page 8). In addition, the correct percentage of females ranged from 11% (Santo et al, 2018) to 89% (Hammonds et al, 2015), instead of 24% to 57.4% as presented in the text (line 49, page 8). If possible,
---

	please add the mean age and number/percentage of females for the total sample of each study in Table 1 and present these total sample data in the text instead of the ranges from intervention and control groups. Furthermore, in the text, it is stated that one study reported adherence according to pill count (lines 3-5, page 9); however, in Table 1, it reported that 2 studies had pill count as the outcome measure (Hammonds et al, 2015 and Labovitz et al, 2017). Please report the correct numbers in the text. c) There is also incorrect information extracted from the original study articles, as follows. Please check.  • The sample size reported in Table 1 should be the number of patients randomised in the studies, instead of number of patients included in the final analysis. As such, the sample size numbers are incorrect in 6 studies, as follows: i) Hammonds et al, 2015 should be 57 instead of 40, ii) Labovitz et al, 2017 should be 28 instead of 27, iii) Lakshminarayana et al, 2017 should be 215 instead of 158, iv) Mira et al, 2014 should be 102 instead of 99, v) Santo et al, 2018 should be 163 instead of 101 and, vi) Svendsen et al, 2018 should be 134 instead of 120. • In Santo et al, 2018, data on mean age presented in Table 1 are incorrect. As published in the paper, data for the intervention group and control group are 58.4 (SD 9.04) and 56.8 (8.64), respectively. In addition, the health problem of target population should be coronary heart disease instead of cardiovascular disease. • In Shah et al, 2016, as published in the paper, the medication adherence was measured using medication possession ratio. There is no mention to the assessment of adherence using the MMAS, as presented in Table 1. Therefore, please remove the sentence 'with one of these also access to the prescribe medication' from lines 54-56 in page 8 and change 'four' to 'three' studies that measured adherence using the 8-item MMAS. • In Svendsen et al, 2018, as published in the paper, the follow-up interval was 28 days instead of 38 days, which is presented in Table 1. d) The 3rd paragraph of page 8 (lines 52-59, page 8, continuing on lines 3-8, page 9), regarding the measures of medication adherence, should be under the sub-title 'Outcome measurement methods'. e) In the 2nd paragraph of page 9, regarding the health program of the target population, stroke and hypertension should be included under cardiovascular diseases. f) Reference 29 (line 30, page 9) is incorrect, as it should be Svendsen et al, 2018 instead of Svenson et al, 2008. Please check. g) Under the sub-title 'Meta-analysis', please correct the number of the total number of participants based on the correct numbers of the total sample size of each study. h) There are concerns of the results of the meta-analysis as there is also incorrect information in the raw data extracted from the original study articles, as presented in eTable 2. Please see below and check the data.  • Morawski et al, 2018: As published in the paper, the intervention group N=209 instead of 210; control group N=202 instead of 90 and mean (SD) = 5.7 (1.8) instead of 5.74 (1.53) • Santo et al, 2018: As published in the paper, the intervention group mean = 7.11 and N=101. • Shah et al, 2016: MMAS-8 data is used as the adherence measurement used in the analysis; however, MMAS-8 results are not presented in the paper. Were these results obtained from the authors via email? If so, please make it clear in the text. i) Sub-heading 'Behaviour Change Techniques': The results of the meta-regression presented in the abstract and discussed in the
--	--

	discussion are not presented in the main results text. Please add. j) Sub-heading 'Risk of bias at primary study level':  • In Table 2, 'Blinding of participants and personnel' should be presented in a different column to 'Objective outcome measure', as according to the Cochrane Risk of Bias Tool, the first one is related to 'Performance bias' and the latter one is related to 'Detection bias'. Please also add legends to the colours in the table. • There are inconsistencies in the information on the type of adherence outcome measure presented on the 2nd paragraph (lines 45-52, page 10) compared to Table 1 and eTable 2. For example, in this paragraph, it is stated that 'patient-entered medication logs in an app' was used as the adherence outcome in Hammonds et al, 2015; while, in Table 1, it is stated as pill count. These are different measures, please clarify. Also, in this paragraph, it is stated that medication possession ratio was used as the adherence outcome in Shah et al, 2016; while, in eTable 2, it is stated that the MMAS-8 data was used in the meta-analysis. Please also clarify. k) In the flowchart, in the reasons of full-text articles exclusion, it is stated that 9 articles were excluded for 'wrong outcome measure', 2 for 'outcome data not suitable for meta-analysis', 13 for 'intervention conditions not eligible', 6 for 'control conditions not eligible' and 2 for 'population not eligible'. It is not clear what is meant by wrong outcome measure, please clarify in the text. It is also not clear how the outcome data was not suitable for meta-analysis, please also clarify. In addition, it would be interesting to know why 13 studies were excluded for intervention conditions not eligible. Was it because the intervention was not an app-based intervention? If so, I would suggest changing the label to "Not an app-based intervention", making it clearer. Similarly, it would be good to know the reason why 6 studies were excluded for control conditions not eligible. Finally, clarifying why in 2 studies the population was not eligible is important, as in the methods it is stated that the review included studies with participants of any age and medical condition. 3) Discussion: Please include a paragraph comparing the results of this review with other previous similar reviews, if any. Minor changes:  1) Change 'out with' to 'in addition to the' in line 48 of page 6. 2) A full-stop is missing on line 42 of page 8 after 2014-2018. 3) Add 'adherence using' after 'four measured' on line 54 of page 8. 4) Please correct 'Moriksy' to 'Morisky' on line 54 of page 8. 5) Table 1:  a) Please change % female under Intervention group to Female, N(%). b) Please correct 'Labowitz et al' to Labovitz et al'. c) Please change MMAS 4 to 4-item MMAS. d) Please also add Self-report before 8-item MMAS and 4-item MMAS, as it is done for the A14 scale.
--	---

REVIEWER	James X. Zhang Department of Medicine The University of Chicago 5841 S Maryland Ave., MC 5000 Chicago, IL 60637 USA
REVIEW RETURNED	16-Sep-2019

GENERAL COMMENTS	Medication non-adherence is a persistent challenge and an elusive issue in healthcare around the globe. The authors aimed to estimate the
---

	effectiveness of app-based interventions designed to support medication adherence and establish which behaviour change techniques utilised by applications are associated with an adherence effect. The study is topically important and the approach is in general appropriate. 1. Causes for medication non-adherence. It is well established that medication non-adherence can be due to a wide variety of reasons, including cultural factors, individual preferences, side effects, and/or financial barriers. While the authors cited a number of studies, there is little or no discussion of those causes for medication non-adherence. Hence it's unclear how the simple utilization of mobile devices can be effective in reducing medication non-adherence caused by many factors. In fact, a recent study published in the JAMA suggested that the low-cost reminder devices did not improve adherence among nonadherent patients who were taking up to 3 medications to treat common chronic conditions (https://jamanetwork.com/journals/jamainternalmedicine/fullarticle/2605527.) Some elaboration on what aspect(s) of medication non-adherence those mobile devices were used to address and the heterogeneity among those devices will help readers to understand why the finding of the study is such that those mobile devices are effective. 2. Heterogeneity among patient populations. Clearly, the patient populations reported in the included studies were very different across countries, with each having unique patient characteristics and country healthcare system characteristics. Even within the same country, the patients may differ in terms of their disease burden, and hence their non-adherence behaviors. Some discussion in this aspect of heterogeneity will help readers to understand if the included studies can be generalized given the significantly different patient populations and varied reasons for medication non-adherence. 3. Outcome measure. As the authors properly pointed out, the outcome measures included in some studies lack objectivity. In fact, some of the measures barely qualify as a good, robust measure of medication non-adherence. How was it consolidated and measured in the meta analysis? A more nuanced description of the analytic approach will help readers to assess the reliability and generalizability of this study. 4. Small sample size. With 9 studies included with a total of 1005 participants, the average N is only 111. Are these all non-adherent patients? Please clarify. With a wide range of risk factors for non-adherence, high level of heterogeneity among patient populations, and small sample size for the studies, it may pose a significant challenge to the validity of this study. The authors should at least discuss in greater details the nature of those mobile devices and how they can perceivably change the non-adherence behaviors, and caution the generalizability.
--	---

VERSION 1 – AUTHOR RESPONSE

Reviewer: 1

Reviewer Name

Lauri Linder

Institution and Country

University of Utah
United States of America

Please state any competing interests or state 'None declared':

None

Please leave your comments for the authors below

Thank you for the opportunity to review this manuscript. As apps to support health behaviors such as medication adherence become more ubiquitous, the need to evaluate their efficacy is relevant.

Given the heterogeneity of the studies in relation to the study sample size and characteristics, approach measuring adherence, and follow up period, I do have some cautions regarding the strength of the claims that can be made regarding the efficacy of apps.

Reviewer Comment:

In the methods section, please address the inclusion/exclusion criteria earlier within this section and state these criteria more explicitly, including if you sought to include published articles only.

Response to reviewer:

Many thanks for your comment. We have now added this clarity in the methods paragraph titled 'Study Selection'.

Reviewer Comment:

The inclusion criteria speak to inclusion of individuals of any age who were taking one or more prescribed medications. Did these prescribed medications need to be taken on a given schedule or were "as needed" prescribed medications regarded as meeting this eligibility criterion? If so, did the studies adequately account for adherence with "as needed" medications in addition to those scheduled, prescribed medications?

Response to reviewer:

Many thanks for your comment. We did not stipulate whether the intervention needed to target at 'as needed' or 'scheduled' medications in order to be eligible.

Reviewer Comment:

How, specifically, did you define "app-based intervention" for the sake of your review? Is an app that delivers an alert one that meets criteria? In the discussion section, you speak of patients who "participate in medication adherence interventions delivered by mobile applications." Please provide a clear conceptual definition of this inclusion criterion and how you operationalized it across studies deemed as meeting criteria.

Response to reviewer:

Thank you for this important suggestion for improvement. We have elaborated on the inclusion and exclusion criteria accordingly.

I am also unclear as to how "behavior change technique" was defined for the sake of this review. Again, is a simple reminder alert system sufficient to meet this criterion? Please state clearly. I also do not see sufficient evidence from the manuscript to feel confident that any of these interventions were "tailored." The NCI defines a tailored intervention as "The use of communication, drugs, or other types of treatments that are specific for an individual or a group to improve health or change behavior." If some of the interventions are ones that meet the generally accepted definition of a tailored intervention, please include sufficient supporting evidence to support this claim. Perhaps Table 1 could be expanded to include these descriptions.

Response to reviewer:

Thank you for this important suggestion for improvement. We have now included two additional supplementary tables: the first shows all behaviour change techniques coded for each of the included studies. The second provides definitions of each of the behaviour change techniques that were coded and gives examples of 2 studies in which each of the behaviour change techniques were present.

Reviewer Comment:

The first paragraph of the discussion seems better suited to the results section. As I stated earlier, given the heterogeneity of the studies included in this review, I am not fully confident that the authors can claim that those individuals using medication reminder apps are more likely to adhere to their medications. Perhaps the better claim is to indicate that they report higher adherence given the predominance of self-report vs. objective measures to ascertain adherence within this group of studies.

Response to reviewer:

Thank you, we have highlighted in the discussion and conclusion sections that the result of the meta-analysis should be interpreted cautiously owing to the fact that data from 6 of the 9 studies included in the meta-analysis was self-reported. We have changed the wording of the result of the meta-analysis of the 5 studies which reported data using the Morisky Adherence Scale to read "patients using medication adherence interventions delivered by mobile applications report higher adherence to prescribed medications."

Reviewer Comment:

General comments:

- Please avoid phrases such as "To our knowledge, this is the first ...," just state what the project adds to the larger body of knowledge.

Response to reviewer:

Thank you, we have taken this comment on board and adjusted language accordingly.

- In the final paragraph on page 8, lines 52-59, please review how you are using the terms "measured" and "measuring." The studies are measuring adherence rather than the scale

Response to reviewer:

Thank you, we have amended the language accordingly.

Reviewer: 2

Reviewer Name

leslie saxon

Institution and Country

University of Southern California

USA

Please state any competing interests or state 'None declared':

None

Please leave your comments for the authors below

In this study, based on a thorough literature review spanning 4 years, nine randomized studies of medication adherence, that utilized various app-based methods for tracking adherence were combined using meta-analysis to determine effectiveness. The study found that Apps were 2 times more likely to drive adherence, compared to control groups.

In the introduction, the authors appropriately note the adverse effects on disease outcomes as well as the economic burden of medication non-adherence in patients across the spectrum of chronic disease. They also note that despite the presence of thousands of medication adherence apps designed all over the globe, there has been very little rigorous study of their effectiveness. Also, the authors had to review 9000 studies to find 9 in the published literature that conformed to these criteria.

Of the nine studies reported, most assessed adherence using self-reporting, without clear validation of drug ingestion.

Nonetheless, the analysis points to the importance and promise of providing comprehensive tools for patient self-management of chronic conditions. Drug adherence is just one aspect of digitally enabled patient self-management. Education and diagnostic tools are another. Medical care communication streams, medication access, nimble virtual care paradigms that don't require a patient and physician to be in the same room are also critical.

I think that the strength of this analysis is that it points to the need for more rigorous and comprehensive study medical applications for disease management. Outcome data is critical to identifying key elements of digital management that can scale across disease states and across the globe.

Notes for Specific Edits

Reviewer Comment:

1. The paper needs to be shortened, both in terms of tables and text.

Response to reviewer:

Thank you for your comment. We have removed areas of repetition and shortened the language used where possible. We have however added further detail in some places to meet requests of the other reviewers.

Reviewer Comment:

2. The authors should review behavioral aspects of top-rated drug adherence apps from the app stores, this is important because these are the apps that patients are using

Response to reviewer:

Many thanks for your comment. Our study aim was to answer the question as to whether or not apps designed to support medication adherence are effective and bring about a difference in medication adherence. Therefore we included studies of effectiveness in this review and meta-analysis and we believe it wouldn't be appropriate to consider apps which have not been rigorously evaluated through a study which has been published and peer reviewed when aiming to answer a question regarding effectiveness. Including such apps would of course also serve to lengthen both the tables and text of the manuscript.

Reviewer Comment:

3. The authors should review FDA approved commercial digital pharma products aimed at adherence, like Proteus Digital Health.

Response to reviewer:

Many thanks for your suggestion. This is beyond the scope of the research question for this review and meta-analysis. We sought to include mobile interventions which have been tested for efficacy in a

trial which has been published in a peer reviewed journal.

Reviewer: 3

Reviewer Name

Karla Santo

Institution and Country

Hospital Israelita Albert Einstein, São Paulo, Brazil

Please state any competing interests or state 'None declared':

None declared

Please leave your comments for the authors below

The authors describe a systematic review with meta-analysis that aimed to address an important research question of whether mobile apps are effective in improving medication adherence. In addition, the authors aimed to identify the apps characteristics, its relationship with behavioural change techniques and whether those influenced effectiveness. As mobile apps have been increasingly used in healthcare and, more recently, as a potential tool to improve adherence, this review provides relevant data to support the use of such apps in clinical practice. The paper is well-written, and the review was performed in a proper scientific manner according to the PRISMA guidelines. Of note, the review was registered in the PROSPERO register, had a robust search strategy, and had 2 reviewers independently reviewing the abstracts and full text articles, extracting the data and assessing the included articles for the risk of bias. However, there are concerns about the results presented and a few points in the methods that need clarification, as listed below.

1) Methods - Study selection:

Reviewer Comment:

In the eligibility criteria, it is stated that the study needed to have an intervention group which received an app-based intervention to help, support or advise about medication adherence. It is unclear whether studies that have a mobile app as one component of a broader intervention addressing medication adherence could be included in the review, for example an intervention that included nurse-led counselling about medication adherence in addition to the mobile app, or whether the mobile app should be the only component of the intervention. Please clarify in the text.

Response to reviewer:

Thank you, we agree this is a helpful point to clarify and have added this to the eligibility criteria under the Methods subheading 'Study Selection'

2) Results:

Reviewer Comment:

a) There are inconsistencies in the information presented in the text and in the flowchart. In the text, it is reported that the database search identified 13,295 articles; however, in the flowchart, it is stated that 13,259 records were identified through database searching. Similarly, in the text, it is reported that 80 articles were eligible for full-text review; however, in the flowchart, it is stated that 83 full-text articles were assessed for eligibility. Please report the correct numbers in the text.

Response to reviewer:

Thank you for noting these typing errors. These have now been addressed in the main text

Reviewer Comment:

b) There also several inconsistencies between the information in the text and in Table 1. For example, in Table 1, the mean age ranged from 20.3 (Hammonds et al, 2015) to 73.8 (Mertens et al, 2016), instead of 20.3 to 70.9 as presented in the text (lines 42-44, page 8). In addition, the correct percentage of females ranged from 11% (Santo et al, 2018) to 89% (Hammonds et al, 2015), instead of 24% to 57.4% as presented in the text (line 49, page 8). If possible, please add the mean age and number/percentage of females for the total sample of each study in Table 1 and present these total sample data in the text instead of the ranges from intervention and control groups. Furthermore, in the text, it is stated that one study reported adherence according to pill count (lines 3-5, page 9); however, in Table 1, it reported that 2 studies had pill count as the outcome measure (Hammonds et al, 2015 and Labovitz et al, 2017). Please report the correct numbers in the text.

Response to reviewer:

Thank you for noticing these points. We have now addressed these in the main text.

Reviewer Comment:

c) There is also incorrect information extracted from the original study articles, as follows. Please check.

- The sample size reported in Table 1 should be the number of patients randomised in the studies, instead of number of patients included in the final analysis. As such, the sample size numbers are incorrect in 6 studies, as follows: i) Hammonds et al, 2015 should be 57 instead of 40, ii) Labovitz et al, 2017 should be 28 instead of 27, iii) Lakshminarayana et al, 2017 should be 215 instead of 158, iv) Mira et al, 2014 should be 102 instead of 99, v) Santo et al, 2018 should be 163 instead of 101 and, vi) Svendsen et al, 2018 should be 134 instead of 120.

Response to reviewer:

Thank you, we have updated the table to reflect the recruited sample size rather than the sample number included in the meta-analysis.

- In Santo et al, 2018, data on mean age presented in Table 1 are incorrect. As published in the paper, data for the intervention group and control group are 58.4 (SD 9.04) and 56.8 (8.64), respectively. In addition, the health problem of target population should be coronary heart disease instead of cardiovascular disease.

Response to reviewer:

Thank you, we have amended the details for your paper accordingly.

- In Shah et al, 2016, as published in the paper, the medication adherence was measured using medication possession ratio. There is no mention to the assessment of adherence using the MMAS, as presented in Table 1. Therefore, please remove the sentence 'with one of these also access to the prescribe medication' from lines 54-56 in page 8 and change 'four' to 'three' studies that measured adherence using the 8-item MMAS.

Response to reviewer:

Shah et al published the Morisky adherence data in a supplementary file. We included this data in the meta-analysis and this was most comparable to the published data for the other included studies.

- In Svendsen et al, 2018, as published in the paper, the follow-up interval was 28 days instead of 38 days, which is presented in Table 1.

Response to reviewer:

Thank you for noticing this typo. We have made the required amendment.

d) The 3rd paragraph of page 8 (lines 52-59, page 8, continuing on lines 3-8, page 9), regarding the measures of medication adherence, should be under the sub-title 'Outcome measurement methods'.

Response to reviewer:

Thank you, we agree and we have removed some of the repetition of detail.

e) In the 2nd paragraph of page 9, regarding the health program of the target population, stroke and hypertension should be included under cardiovascular diseases.

Response to reviewer:

We have amended accordingly.

f) Reference 29 (line 30, page 9) is incorrect, as it should be Svendsen et al, 2018 instead of Svenson et al, 2008. Please check.

Response to reviewer:

Thank you, we had also spotted this error since submission and have corrected this.

g) Under the sub-title 'Meta-analysis', please correct the number of the total number of participants based on the correct numbers of the total sample size of each study.

Response to reviewer:

We only included the number of participants for whom data was available in the meta-analysis. We have now amended this sentence to make this clearer.

h) There are concerns of the results of the meta-analysis as there is also incorrect information in the raw data extracted from the original study articles, as presented in eTable 2. Please see below and check the data.

- Morawski et al, 2018: As published in the paper, the intervention group N=209 instead of 210; control group N=202 instead of 90 and mean (SD) = 5.7 (1.8) instead of 5.74 (1.53)

Response to reviewer:

Thank you. Both of these figures for Morawski et al's paper were transcribing errors and the sample sizes of 209 and 202 were used for the intervention and control group respectively for the performance of the meta-analysis.

- Santo et al, 2018: As published in the paper, the intervention group mean = 7.11 and N=101.

Response to reviewer:

Thank you, for your comment. For your paper, we included data relating only to the advanced application that was tested as this was most comparable to the interventions in the other studies included in the meta-analysis, with the basic app providing a daily reminder only. We have now added detail to the main text of the manuscript to make this clearer, under the Methods subsection entitled 'Data Extraction and Synthesis'.

- Shah et al, 2016: MMAS-8 data is used as the adherence measurement used in the analysis; however, MMAS-8 results are not presented in the paper. Were these results obtained from the authors via email? If so, please make it clear in the text.

Response to reviewer:

These were published in the Supplemental Material file. We have now made this clearer in the main text.

i) Sub-heading 'Behaviour Change Techniques': The results of the meta-regression presented in the abstract and discussed in the discussion are not presented in the main results text. Please add.

Response to reviewer:

Thank you, we have added this under the Results sub-header 'Behaviour Change Techniques'.

j) Sub-heading 'Risk of bias at primary study level':

- In Table 2, 'Blinding of participants and personnel' should be presented in a different column to 'Objective outcome measure', as according to the Cochrane Risk of Bias Tool, the first one is related to 'Performance bias' and the latter one is related to 'Detection bias'. Please also add legends to the colours in the table.

Response to reviewer:

We have taken this comment on board and adjusted the table accordingly.

- There are inconsistencies in the information on the type of adherence outcome measure presented on the 2nd paragraph (lines 45-52, page 10) compared to Table 1 and eTable 2. For example, in this paragraph, it is stated that 'patient-entered medication logs in an app' was used as the adherence outcome in Hammonds et al, 2015; while, in Table 1, it is stated as pill count. These are different measures, please clarify.

Response to reviewer:

Thank you, we have amended table 1 accordingly.

Also, in this paragraph, it is stated that medication possession ratio was used as the adherence outcome in Shah et al, 2016; while, in eTable 2, it is stated that the MMAS-8 data was used in the meta-analysis. Please also clarify.

Response to reviewer:

Thank you, we have amended this error to correctly reflect that the MMAS-8 data was included in the meta-analysis for Shah et al.

k) In the flowchart, in the reasons of full-text articles exclusion, it is stated that 9 articles were excluded for 'wrong outcome measure', 2 for 'outcome data not suitable for meta-analysis', 13 for 'intervention conditions not eligible', 6 for 'control conditions not eligible' and 2 for 'population not eligible'. It is not clear what is meant by wrong outcome measure, please clarify in the text. It is also not clear how the outcome data was not suitable for meta-analysis, please also clarify. In addition, it would be interesting to know why 13 studies were excluded for intervention conditions not eligible. Was it because the intervention was not an app-based intervention? If so, I would suggest changing the label to "Not an app-based intervention", making it clearer. Similarly, it would be good to know the reason why 6 studies were excluded for control conditions not eligible. Finally, clarifying why in 2 studies the population was not eligible is important, as in the methods it is stated that the review included studies with participants of any age and medical condition.

Response to reviewer:

Thank you: Outcome data were not suitable for inclusion in the meta-analysis where the intervention and outcome measure related to other aspects of self-management as well as medication adherence and data on medication adherence alone could not be provided. Intervention conditions were not eligible where the application was only one component of a complex, multicomponent intervention. The control conditions were not eligible where the control group received another form of adherence intervention (e.g. a paper based intervention, or a different version of an application. The population was not eligible if they were healthy recruits who were requested to adhere to a placebo medication. We have now added this detail to the inclusion and exclusion criteria.

3) Discussion: Please include a paragraph comparing the results of this review with other previous similar reviews, if any.

Minor changes:

1) Change 'out with' to 'in addition to the' in line 48 of page 6.

Response to reviewer.

Thank you. We have changed the wording to 'other than' as studies were eligible for inclusion even if other components of the treatment regimen were measured, but only if they specifically reported adherence to medication as an outcome.

2) A full-stop is missing on line 42 of page 8 after 2014-2018.

Response to reviewer:

Thank you.

3) Add 'adherence using' after 'four measured' on line 54 of page 8.

Response to reviewer:

Thank you.

4) Please correct 'Moriksy' to 'Morisky' on line 54 of page 8.

Response to reviewer:

Thank you.

5) Table 1:

a) Please change % female under Intervention group to Female, N(%).

Response to reviewer:

Thank you.

b) Please correct 'Labowitz et al' to Labovitz et al'.

Response to reviewer:

Thank you. We noticed we have made the same error in Table 2 and have also corrected this.

c) Please change MMAS 4 to 4-item MMAS.

Response to reviewer:

Thank you.

d) Please also add Self-report before 8-item MMAS and 4-item MMAS, as it is done for the A14 scale.

Response to reviewer:

Thank you.

Reviewer: 4

Reviewer Name

James X. Zhang

Institution and Country

Department of Medicine
The University of Chicago
5841 S Maryland Ave., MC 5000
Chicago, IL 60637
USA

Please state any competing interests or state 'None declared':

None

Please leave your comments for the authors below

Medication non-adherence is a persistent challenge and an elusive issue in healthcare around the globe. The authors aimed to estimate the effectiveness of app-based interventions designed to support medication adherence and establish which behaviour change techniques utilised by applications are associated with an adherence effect. The study is topically important and the approach is in general appropriate.

Reviewer Comment:

1. Causes for medication non-adherence. It is well established that medication non-adherence can be due to a wide variety of reasons, including cultural factors, individual preferences, side effects, and/or financial barriers. While the authors cited a number of studies, there is little or no discussion of those causes for medication non-adherence. Hence it's unclear how the simple utilization of mobile devices can be effective in reducing medication non-adherence caused by many factors. In fact, a recent study published in the JAMA suggested that the low-cost reminder devices did not improve adherence among nonadherent patients who were taking up to 3 medications to treat common chronic conditions (<https://jamanetwork.com/journals/jamainternalmedicine/fullarticle/2605527>.) Some elaboration on what aspect(s) of medication non-adherence those mobile devices were used to address and the heterogeneity among those devices will help readers to understand why the finding of the study is such that those mobile devices are effective.

Response to reviewer:

Thank you for this comment. This has prompted us to include a summary table of the Behaviour Change Techniques utilised by each of the mobile device applications as a supplemental table and a further table with the definitions of these coded techniques and examples. This illustrates some of the

variance observed between the design of the application interventions.

Reviewer Comment:

2. Heterogeneity among patient populations. Clearly, the patient populations reported in the included studies were very different across countries, with each having unique patient characteristics and country healthcare system characteristics. Even within the same country, the patients may differ in terms of their disease burden, and hence their non-adherence behaviors. Some discussion in this aspect of heterogeneity will help readers to understand if the included studies can be generalized given the significantly different patient populations and varied reasons for medication non-adherence.

Response to reviewer:

Many thanks for your comment. We agree these are helpful points for discussion and have added these to our discussion regarding potential causes for variance in the effect size seen between studies. This can be found in the final paragraph of the Discussion sub-header entitled 'Principle Findings'.

Reviewer Comment:

3. Outcome measure. As the authors properly pointed out, the outcome measures included in some studies lack objectivity. In fact, some of the measures barely qualify as a good, robust measure of medication non-adherence. How was it consolidated and measured in the meta analysis? A more nuanced description of the analytic approach will help readers to assess the reliability and generalizability of this study.

Response to reviewer:

Thank you for this suggestion. We agree that the outcome measures varied across the studies and the use of self-report measures is a limitation of both the primary studies and the review. To performed two meta-analyses; the first pooling data from all studies, and the second pooling data from studies which used the Morisky medication adherence scale to investigate whether there was a difference in the result when studies which used a different outcome measure were removed. In both instances we used the standardised mean difference between the intervention and control groups. We have included the raw data we used for the meta-analysis in the supplementary information file and have now provided more detail on the meta-analysis in the main text.

Reviewer Comment:

4. Small sample size. With 9 studies included with a total of 1005 participants, the average N is only 111. Are these all non-adherent patients? Please clarify. With a wide range of risk factors for non-adherence, high level of heterogeneity among patient populations, and small sample size for the studies, it may pose a significant challenge to the validity of this study. The authors should at least discuss in greater details the nature of those mobile devices and how they can perceivably change the non-adherence behaviors, and caution the generalizability.

Response to reviewer:

Thank you for your suggestion. We have elaborated on the reasons for caution in interpreting or generalising the results of this study in the conclusion. We have also added a short paragraph at the end of the Discussion subheading entitled 'Principle Findings', stating that the heterogeneity between studies and variance in effect size mean interpretation and generalisation of the results is necessarily cautious.

VERSION 2 – REVIEW

REVIEWER	Lauri Linder University of Utah USA
-----------------	---

REVIEW RETURNED	22-Oct-2019
GENERAL COMMENTS	Overall, the authors have been receptive to previous feedback and have worked to adjust content to result in claims that are more appropriate given the current state of the science. This manuscript also provides guidance for future needed work to evaluate the efficacy of medication reminder apps in future studies.
REVIEWER	Karla Santo Hospital Israelita Albert Einstein, São Paulo, Brazil
REVIEW RETURNED	22-Oct-2019
GENERAL COMMENTS	I thank the authors for the responses to the reviewer comments. The authors have clarified most points raised by the reviewers satisfactorily. However, there is a one major point about the inclusion and exclusion criteria that still needs to be clarified and a few minor points that need to be addressed. Major point: 1) Methods – Study Selection (pages 7 and 8): One of the inclusion criteria states ‘a comparator group which received usual care, another medication adherence intervention that did not utilise an app or an app which did not include any behaviour change techniques’. Importantly, the definition of the comparator in the PROSPERO register is different, stating that the comparator group could be either: treatment as usual, no intervention or another medication adherence intervention of any kind. However, one of the exclusion criteria in the manuscript states that a study would be excluded if ‘comparator group received an adherence intervention in another format, such as paper-based or different application’. It seems that these criteria are conflicting. Could the comparator be another medication intervention or not? Please clarify which types of comparators were able to be included in this review and modify the text to make it clearer. Minor points: 2) Abstract – Setting (page 2): In the abstract, under ‘Setting’, it is stated that the comparator is usual care; however, in the main text, under Methods - ‘Study Selection’, it is also mentioned that the comparator could be another medication adherence intervention which did not utilise an app, or an app which did not include any behaviour change techniques. Please amend the abstract with the complete information. 3) Abstract – Intervention (page 2) and Main text – Methods (Study Selection) (page 7): I believe the correct term is ‘personal digital assistant’ instead of ‘personal desktop assistant’. Please check and correct, if necessary. 4) Abstract – Primary and Secondary Outcome Measures (page 2): Please revise the second sentence about the secondary outcome, as the outcome should not contain an action verb in it. 5) Abstract – Results (page 3): Given the point made in comment 1), I would suggest changing ‘usual care’ to ‘comparator’, as usual care was not the only possible comparator in the studies. 6) Methods – Study Selection (page 7): Under Inclusion criteria (i), please mirror the text used in the Abstract to ‘for any health condition for any duration’. 7) Results – Flowchart diagram (page 27): In relation to a previous comment made on the reasons of full-text exclusion provided in the

	Flowchart diagram, I thank the authors for explaining these reasons in their response letter and adding more information in the inclusion/exclusion criteria section. However, it is still unclear what was defined as 'wrong outcome measure'. In addition, the labels provided in the figure are still unclear. I would suggest changing some of the labels to make them more specific, e.g. 'Complex interventions' instead of 'intervention conditions not eligible', 'adherence outcome data not available' instead of 'outcome data not available', 'not an app-based intervention' instead of 'intervention conditions not eligible', 'Healthy adults' population' instead of 'population not eligible', etc. The figure should be self-explanatory that readers shouldn't need to go to the text to understand the content of the figure. 8) Results (page 10): It would be nice to have a few statistics that summarise the characteristics of the included studies in the text, including the overall mean or median sample size of the included studies, as well as the sample size range; the overall mean or median of follow-up period, as well as the range; the overall mean age and mean % females among the included studies. 9) Discussion: There was no response in the authors' response letter in relation to the following previous comment 'Please include a paragraph comparing the results of this review with other previous reviews, if any.' Minor editing comments: 1) Throughout the abstract and main text: Please standardize the use of the terms 'application' and 'app', by spelling it out on the first use and using the abbreviated form thereafter. 2) Throughout the abstract and main text: Please revise and standardize the use of the terms 'effectiveness' and 'efficacy' as they are being used interchangeably, when they have different meanings. 3) Throughout the abstract and main text: Please standardize the use of the terms 'behaviour change techniques' and 'BCTs', by spelling it out on the first use and using the abbreviated form thereafter. 4) Abstract – Objectives (page 2): Please start the sentence with 'To estimate'. 5) Introduction (page 5): a. Please add the word 'to' between the words 'estimated' and 'exceed' on line 16. b. Please add a full stop after the word 'lower' on line 35. c. Please remove the comma after the word 'meta-analysis' on line 56. 6) Methods – Statistical analysis (page 9): a. Please correct 'was' to 'were' on line 35. b. Please remove the words 'was used' on line 55, as they are repeated in the sentence. 7) Results (page 10): Please put a comma after duplicates on line 12. 8) Results – Outcome Measurement Methods (page 10): Please change 'by' to 'in' on line 46.
--	--

REVIEWER	James X. Zhang The University of Chicago USA
REVIEW RETURNED	23-Oct-2019

GENERAL COMMENTS

This is a much-improved manuscript with tighter conclusion and rigorous discussion of the study limitation and caution warranted.

VERSION 2 – AUTHOR RESPONSE

Reviewer: 1

Reviewer Name

Lauri Linder

Institution and Country

University of Utah
USA

Please state any competing interests or state 'None declared':
None declared.

Please leave your comments for the authors below

Overall, the authors have been receptive to previous feedback and have worked to adjust content to result in claims that are more appropriate given the current state of the science. This manuscript also provides guidance for future needed work to evaluate the efficacy of medication reminder apps in future studies.

Author response:

Many thanks for your positive feedback

Reviewer: 3

Reviewer Name

Karla Santo

Institution and Country

Hospital Israelita Albert Einstein, São Paulo, Brazil

Please state any competing interests or state 'None declared':
None declared.

Please leave your comments for the authors below

I thank the authors for the responses to the reviewer comments. The authors have clarified most points raised by the reviewers satisfactorily. However, there is a one major point about the inclusion and exclusion criteria that still needs to be clarified and a few minor points that need to be addressed.

Author response:

Thank you for your careful consideration of our manuscript and detailed suggestions to help improve our manuscript. We hope we have now addressed these in full as follows.

Major point:

1) Methods – Study Selection (pages 7 and 8): One of the inclusion criteria states ‘a comparator group which received usual care, another medication adherence intervention that did not utilise an app or an app which did not include any behaviour change techniques’. Importantly, the definition of the comparator in the PROSPERO register is different, stating that the comparator group could be either: treatment as usual, no intervention or another medication adherence intervention of any kind. However, one of the exclusion criteria in the manuscript states that a study would be excluded if ‘comparator group received an adherence intervention in another format, such as paper-based or different application’. It seems that these criteria are conflicting. Could the comparator be another medication intervention or not? Please clarify which types of comparators were able to be included in this review and modify the text to make it clearer.

Author response:

Thank you, our apologies that this has not been clearer until now. We have applied to PROSPERO to update our record (a full audit trail of changes will be available to viewers) and have now ensured that the comparator criteria are consistent between the abstract and the main text.

Minor points:

2) Abstract – Setting (page 2): In the abstract, under ‘Setting’, it is stated that the comparator is usual care; however, in the main text, under Methods - ‘Study Selection’, it is also mentioned that the comparator could be another medication adherence intervention which did not utilise an app, or an app which did not include any behaviour change techniques. Please amend the abstract with the complete information.

Author response:

Thank you, we have amended this as you suggest.

3) Abstract – Intervention (page 2) and Main text – Methods (Study Selection) (page 7): I believe the correct term is ‘personal digital assistant’ instead of ‘personal desktop assistant’. Please check and correct, if necessary.

Author response:

Thank you, we have amended this to Personal Digital Assistant

4) Abstract – Primary and Secondary Outcome Measures (page 2): Please revise the second sentence about the secondary outcome, as the outcome should not contain an action verb in it.

Author response:

Thank you, we have amended accordingly.

5) Abstract – Results (page 3): Given the point made in comment 1), I would suggest changing ‘usual care’ to ‘comparator’, as usual care was not the only possible comparator in the studies.

Author response:

Thank you, we have amended accordingly.

6) Methods – Study Selection (page 7): Under Inclusion criteria (i), please mirror the text used in the Abstract to ‘for any health condition for any duration’.

Author response:

Thank you, we have amended accordingly.

7) Results – Flowchart diagram (page 27): In relation to a previous comment made on the reasons of full-text exclusion provided in the Flowchart diagram, I thank the authors for explaining these reasons in their response letter and adding more information in the inclusion/exclusion criteria section. However, it is still unclear what was defined as ‘wrong outcome measure’. In addition, the labels provided in the figure are still unclear. I would suggest changing some of the labels to make them more specific, e.g. ‘Complex interventions’ instead of ‘intervention conditions not eligible’, ‘adherence outcome data not available’ instead of ‘outcome data not available’, ‘not an app-based intervention’ instead of ‘intervention conditions not eligible’, ‘Healthy adults’ population’ instead of ‘population not eligible’, etc. The figure should be self-explanatory that readers shouldn’t need to go to the text to understand the content of the figure.

Author response:

Thank you, we have amended the PRISMA flow accordingly with this detail.

8) Results (page 10): It would be nice to have a few statistics that summarise the characteristics of the included studies in the text, including the overall mean or median sample size of the included studies, as well as the sample size range; the overall mean or median of follow-up period, as well as the range; the overall mean age and mean % females among the included studies.

Author response:

Thank you, we have now added this detail.

9) Discussion: There was no response in the authors’ response letter in relation to the following previous comment ‘Please include a paragraph comparing the results of this review with other previous reviews, if any.’

Author response:

Thank you. We have now added a paragraph in the discussion under the subheading ‘Strengths and Weaknesses at Review level’

Minor editing comments:

1) Throughout the abstract and main text: Please standardize the use of the terms ‘application’ and ‘app’, by spelling it out on the first use and using the abbreviated form thereafter.

Author response:

Thank you, we have taken this on board and amended accordingly.

2) Throughout the abstract and main text: Please revise and standardize the use of the terms ‘effectiveness’ and ‘efficacy’ as they are being used interchangeably, when they have different meanings.

Author response:

Thank you. We believe the most appropriate term is efficacy given that these interventions were tested in a controlled setting and have amended the manuscript accordingly.

3) Throughout the abstract and main text: Please standardize the use of the terms ‘behaviour change techniques’ and ‘BCTs’, by spelling it out on the first use and using the abbreviated form thereafter.

Author response:

Amended accordingly.

4) Abstract – Objectives (page 2): Please start the sentence with ‘To estimate’.

Author response:
Amended accordingly.

5) Introduction (page 5):

a. Please add the word 'to' between the words 'estimated' and 'exceed' on line 16.

Author response:
Amended accordingly.

b. Please add a full stop after the word 'lower' on line 35.

Author response:
Amended accordingly.

c. Please remove the comma after the word 'meta-analysis' on line 56.

Author response:
Amended accordingly.

6) Methods – Statistical analysis (page 9):

a. Please correct 'was' to 'were' on line 35.

Author response:
Amended accordingly.

b. Please remove the words 'was used' on line 55, as they are repeated in the sentence.
Amended accordingly.

7) Results (page 10): Please put a comma after duplicates on line 12.

Author response:
Amended accordingly.

8) Results – Outcome Measurement Methods (page 10): Please change 'by' to 'in' on line 46.

Author response:
Amended accordingly.

Reviewer: 4

Reviewer Name

James X. Zhang

Institution and Country

The University of Chicago
USA

Please state any competing interests or state 'None declared':
None declared

Please leave your comments for the authors below
 This is a much-improved manuscript with tighter conclusion and rigorous discussion of the study limitation and caution warranted.

Author response:
 Many thanks for your positive feedback.

VERSION 3 – REVIEW

REVIEWER	Lauri Linder University of Utah, USA
REVIEW RETURNED	02-Dec-2019

GENERAL COMMENTS	The authors have been responsive to continued feedback, and I have only minor additional edits. Given the current definitions of apps (software designed for a single purpose and performs a single function) vs. applications (software designed to perform a variety of functions), my recommendation would be to use the term "app" exclusively (unless, of course, speaking of an application). Please be careful in using "data" as a plural term - e.g. page 8 - several instances of "data ... was" rather than "data ... were" Page 11 - please adjust the statement indicating "A further study measured the 4-item Morisky ..." to "A further study measured adherence using the 4-item ..." Given the key limitation of relatively few objective measures of adherence within these studies, my recommendation would be to reframe results in terms of patients reporting higher levels of adherence rather than claiming actual higher levels of adherence. Even improving self-reports of adherence suggest that app use is associated with higher levels of engagement with actual medication adherence. The claim of "efficacy in supporting medication adherence" is defensible. I really hesitate to support claims of "more likely to adhere" even with the stated caveat. Simply stating "higher levels of self-reported adherence" is more clearly defensible based on the measurement approaches in these available sources of evidence. Comment regarding eTable 4 - definition column - I am unclear as to why some cells use italicized text and others do not.
---

REVIEWER	Karla Santo Hospital Israelita Albert Einstein
REVIEW RETURNED	13-Dec-2019

GENERAL COMMENTS	I thank the authors for the responses to the reviewers' comments. The authors have now clarified all the remaining points and I have no further comments.
---

VERSION 3 – AUTHOR RESPONSE

Reviewer: 1

Reviewer Name

Lauri Linder

Institution and Country

University of Utah, USA

Please state any competing interests or state 'None declared':

None

Please leave your comments for the authors below

The authors have been responsive to continued feedback, and I have only minor additional edits.

Given the current definitions of apps (software designed for a single purpose and performs a single function) vs. applications (software designed to perform a variety of functions), my recommendation would be to use the term "app" exclusively (unless, of course, speaking of an application).

Author response: Thank you, we have amended the manuscript, supplemental file, figure legends and PRISMA flow-chart accordingly.

Please be careful in using "data" as a plural term - e.g. page 8 - several instances of "data ... was" rather than "data ... were"

Author response: Thank you, we have amended the manuscript accordingly.

Page 11 - please adjust the statement indicating "A further study measured the 4-item Morisky ..." to "A further study measured adherence using the 4-item ..."

Author response: Thank you, we have adjusted the manuscript accordingly.

Given the key limitation of relatively few objective measures of adherence within these studies, my recommendation would be to reframe results in terms of patients reporting higher levels of adherence rather than claiming actual higher levels of adherence. Even improving self-reports of adherence suggest that app use is associated with higher levels of engagement with actual medication adherence. The claim of "efficacy in supporting medication adherence" is defensible. I really hesitate to support claims of "more likely to adhere" even with the stated caveat. Simply stating "higher levels of self-reported adherence" is more clearly defensible based on the measurement approaches in these available sources of evidence.

Author response: Thank you, we have adjusted the abstract and conclusions accordingly.

Comment regarding eTable 4 - definition column - I am unclear as to why some cells use italicized text and others do not.

Author response: Thank you. Italicised text was used to denote text directly quoting the Behaviour Change Taxonomy. This has been amended to use quotation marks for clarity.

Reviewer: 3

Reviewer Name

Karla Santo

Institution and Country

Hospital Israelita Albert Einstein

Please state any competing interests or state 'None declared':

None declared

Please leave your comments for the authors below

I thank the authors for the responses to the reviewers' comments. The authors have now clarified all the remaining points and I have no further comments.

Author response: Thank you.